# Highly efficient non-relativistic Edelstein effect in nodal *p*-wave magnets

Atasi Chakraborty [1] ✉, Anna Birk Hellenes [1], Rodrigo Jaeschke-Ubiergo [1], Tomás Jungwirth [2,3], Libor Šmejkal[1,2,4,5] ✉ & Jairo Sinova [1,6] ✉

The origin and efficiency of charge-to-spin conversion, known as the Edelstein effect (EE), has been typically linked to spin-orbit coupling mechanisms, which require materials with heavy elements within a non-centrosymmetric environment. Here we demonstrate that the high efficiency of spin-charge conversion can be achieved even without spin-orbit coupling in the recently identified coplanar *p*-wave magnets. The non-relativistic Edelstein effect (NREE) in these magnets exhibits a distinct phenomenology compared to the relativistic EE, characterized by a strongly anisotropic response and an out-of-plane polarized spin density resulting from the spin symmetries. We illustrate the NREE through minimal tight-binding models, allowing a direct comparison to different systems. Through first-principles calculations, we further identify the nodal *p*-wave candidate material CeNiAsO as a high-efficiency NREE material, revealing a ~ 25 times larger response than the maximally achieved relativistic EE and other reported NREE in non-collinear magnetic systems with broken time-reversal symmetry. This highlights the potential for efficient spin-charge conversion in *p*-wave magnetic systems.

Conventional spintronics[1,2] typically relies on ferromagnetic materials and external magnetic fields to generate spin-polarized charge currents. More recent spintronic device concepts focus on effects generating spin-current or spin-density accumulation by electric fields[3,4]. The latter is known as the Edelstein effect (EE), appearing in non-centrosymmetric materials and usually originating from relativistic (spin-orbit coupling) effects, as illustrated in Fig. 1a, b[5–7]. Archetypal applications of the EE include spin-orbit torque devices, such as the magneto-electric spin-orbit (MESO) transistor[4,8–10], where efficient charge-to-spin conversion is required to exert a torque that can switch the magnetization orientation[11–17]. The relativistic EE has been demonstrated in numerous systems, including Rashba-Dresselhaus systems[18,19], topological insulators[20], semiconductors[21,22], Weyl semimetals[23,24], oxide interfaces[25–27], 2D electron gases[28,29] and non-centrosymmetric superconductors[30]. The non-equilibrium spin density in these systems originates from spin-orbit coupling (SOC), necessitating heavy elements to reach functional efficiencies of charge-to-spin conversion[26].

The possibility of a non-relativistic EE (NREE) was proposed recently in non-collinear magnetic systems with broken time-reversal symmetry (TRS) band structure and broken parity[31–34], allowing for a current-induced spin accumulation in systems also with possibly light elements. The EE is, in principle, allowed if the system breaks inversion symmetry irrespective of the origin of the response, be it SOC or non-collinear order. The conventional relativistic Rashba-Edelstein effect in TRS systems originates from SOC-induced anti-symmetric spin textures in momentum space. The recently predicted p-wave magnets (Fig. 1c) have TRS as a point group operation, thus exhibiting TRS in momentum space akin to Rashba systems. However, unlike Rashba spin splitting, which is isotropic, our *p*-wave spin splitting is strongly

[1]Institut für Physik, Johannes Gutenberg Universität Mainz, Mainz, Germany. [2]Institute of Physics, Academy of Sciences of the Czech Republic, Cukrovarnická 10, Praha 6, Czech Republic. [3]School of Physics and Astronomy, University of Nottingham, Nottingham, United Kingdom. [4]Max Planck Institute for the Physics of Complex Systems, Nöthnitzer Str. 38, Dresden, Germany. [5]Max Planck Institute for Chemical Physics of Solids, Nöthnitzer Str. 40, Dresden, Germany. [6]Department of Physics, Texas A & M University, College Station, Texas, USA. ✉e-mail: atasi.chakraborty@uni-mainz.de; lsmejkal@pks.mpg.de; sinova@uni-mainz.de

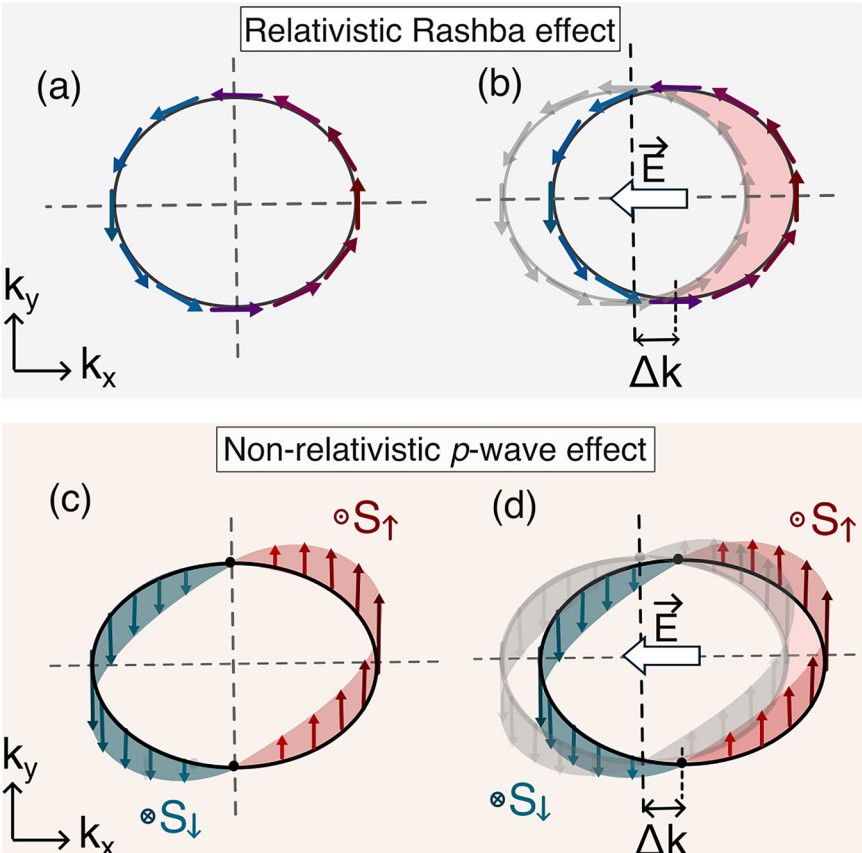

**Fig. 1 | Comparison of charge to spin conversion mechanism for relativistic Rashba and non-relativistic nodal *p*-wave systems. a** Schematic of a relativistic Rashba spin-texture (1 band). **b** (Relativistic Rashba-Edelstein effect) Non-equilibrium redistribution resulting in a net in-plane spin-accumulation. **c** Schematic of non-relativistic out-of-plane spin polarization texture of nodal *p*-wave magnets originating from non-collinear $\mathcal{T}\mathbf{t}$ co-planar magnetic order. **d** (Non-relativistic nodal *p*-wave Edelstein effect) Non-equilibrium out-of-plane spin-accumulation driven by an external electric field.

anisotropic and momentum dependent[35]. This can give rise to out-of-plane non-equilibrium spin accumulation, a feature not permitted in the usual relativistic Rashba-Edelstein effect. This p-wave magnet NREE contrasts with recent reports[33,34] that obtain an NREE in non-collinear magnetic order in non-centrosymmetric crystals, and which do not have TRS as a point group symmetry. In p-wave magnets, the inversion symmetry is broken by the non-collinear coplanar order within a centrosymmetric crystal structure, and TRS in momentum space is preserved even in the presence of SOC.

The prediction and the identification of the favorable characteristics for high NREE efficiency of p-wave magnetic candidates were made possible by applying the spin symmetries approach, instrumental in the discovery of altermagnets[36]. Using the spin symmetries to fully classify and delimit all collinear spin arrangements on crystals leads to the conclusion that only even-parity non-relativistic spin-split band structures (s-, d-, g-, or i-wave) are possible in collinear magnetic systems[36,37]. This narrows the search for *p*-wave (and higher odd-parity) magnets to non-collinear systems[35]. A *p*-wave magnet must break both parity (P) and PT symmetry, and preserve $\mathcal{T}\mathbf{t}$ symmetry, where **t** is a translation. This last symmetry preserves TRS in momentum space by virtue of $\mathcal{T}$ as point group symmetry operation and enforces zero net magnetization. P-wave magnetic candidates with collinear polarized band structure have been identified in the co-planar magnets, which have the spin symmetry [$C_{2\perp}$‖ **t**], where $C_{2\perp}$ being a 180° spin rotation along the axis perpendicular to the spins[35]. This spin symmetry mandates that the spin polarization axis in the electronic structure is perpendicular to the plane of the spins. There exist examples of non-

coplanar *p*-wave magnetic systems which can feature nodeless co-planar anti-symmetric spin polarization. However, for the present study, we are interested in nodal *p*-wave magnets candidates with coplanar spin order in position space showing this unique collinear out-of-plane non-relativistic spin polarization in momentum space[38]. By virtue of $\mathcal{T}$ being an individual symmetry operation of the point group both in the presence and the absence of SOC, the momentum space energy dispersion satisfies the criteria: $E_{\mathbf{k}}(\mathbf{k}, \boldsymbol{\sigma}) = E_{\mathbf{k}}(-\mathbf{k}, -\boldsymbol{\sigma})$, where $\boldsymbol{\sigma}$ is the spin polarization and **k** is the momentum.

This sets the NREE (as shown in Fig. 1d) apart from the usual Rashba EE, whose non-equilibrium spin polarization is on the plane. It also sets the expectation for a large NREE efficiency due to the lack of directional averaging of the spin states contributing to the effect, and the much larger spin-splitting (almost 2 orders of magnitude larger) relative to the relativistic EE[39,40].

We first demonstrate this physics in a simple 2D generic minimal model that exhibits a *p*-wave spin-polarization band structure. We find, even at the simple model level, that the NREE susceptibility of *p*-wave magnets surpasses the magnitude calculated for the relativistic Rashba two-dimensional electron gas (2DEG) and recently reported non-relativistic non-coplanar 3Q anti-ferromagnets (AFMs) with broken TRS[6,33,41]. Extending our model analysis to a bi-Kagome lattice geometry, we find a substantial increase in spin-accumulation density. Finally, through first principle calculations, we identify the *p*-wave candidate CeNiAsO[42–46] showing a highly efficient NREE, exhibiting 25 times larger response than the highest Rashba EE[20] and the non-collinear AFM LuFeO₃[33].

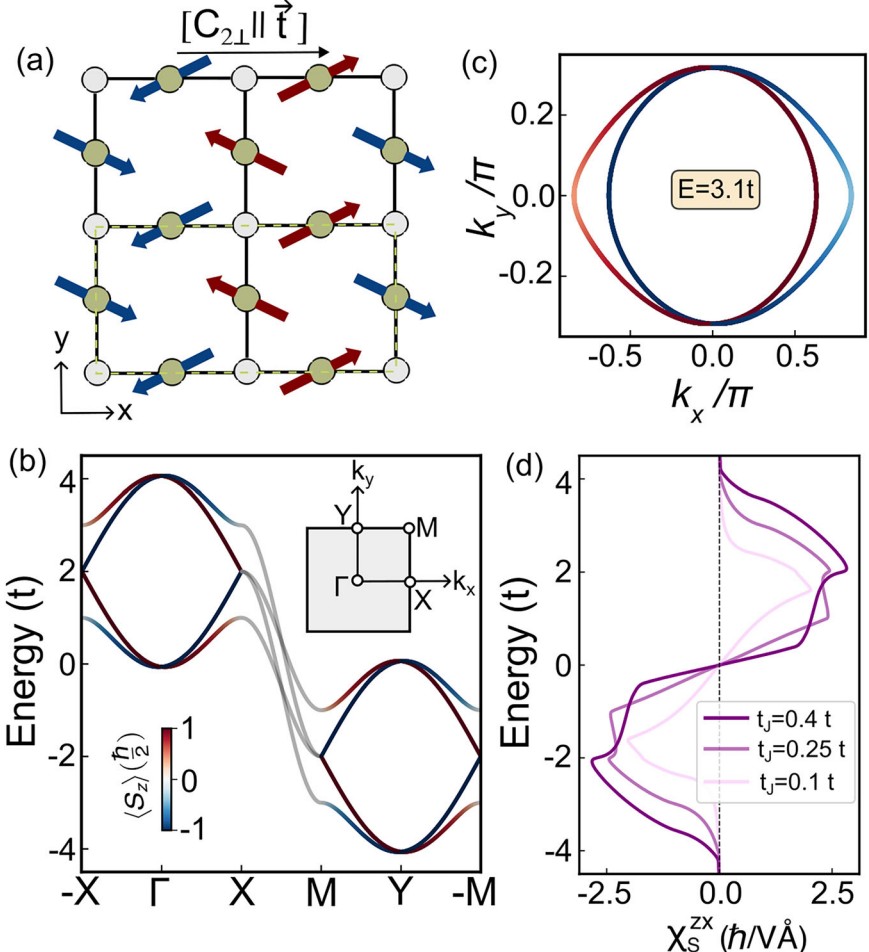

**Fig. 2 | Energy dispersion and Edelstein response for nodal *p*-wave minimal model. a** Schematic of the lattice model with a coplanar non-collinear spin arrangement on the crystal with the spin arrangement that realizes the unconventional p-wave phase. The unit cell is marked with dashed green line. **b** Energy dispersion for $t_J = 0.25\,t$ along high-symmetry momentum paths showing parity broken time-reversal symmetric spin splitting. The color represent the out-of-plane spin component. The opposite spin bands are completely degenerate along the $k_y$ direction forming a nodal line. **c** Constant energy $E = 3.1t$ contour with $t_J = 0.25\,t$. **d** Variation of the NREE response for fixed $\Gamma = 0.01$ eV with the strength of the exchange term. An increase of $t_J$ enhances the shifting of opposite spin bands in opposite momentum directions, therefore, the response strength $\chi_S$ increases.

## Results

### Minimal *p*-wave model

To exemplify the *p*-wave electronic structure and its NREE response, we first formulate a low-energy minimal model for the odd-parity magnets based on the simple tight-binding model presented in Fig. 2a. Here the opposite spins are connected by $[C_{2\perp}\|\mathbf{t}]$ symmetry. The coplanar non-collinear magnetic ordering doubles the unit cell of the square lattice (shown as a dashed green line in Fig. 2a). The electron hopping among the gray nonmagnetic sites is parameterized by $t$, and the p-wave spin splitting originates from the exchange-dependent hopping parametrized by $t_J$. The minimal four-band Hamiltonian is given by[35],

$$H = 2t\left(\cos\frac{k_x}{2}\tau_1 + \cos k_y\right) + 2t_J\left(\sin\frac{k_x}{2}\sigma_1\tau_2 + \cos k_y\sigma_2\tau_3\right). \quad (1)$$

Pauli matrices $\sigma$ and $\tau$ correspond to the spin and site degrees of freedom. The detailed construction of the Hamiltonian is included in Supplementary Fig. 2 and Section 3 of Supplementary Information. Here $t$, $t_J$ represent the spin-independent and exchange-dependent hopping terms between the nearest neighbor sites within the square lattice geometry.

We illustrate the band structure calculated at $t_J = 0.25\,t$ parameter choice of Eq. (1) in Fig. 2b. The fixed energy contour plot at energy $E = 3.1t$ is plotted in Fig. 2c. As it is clear from the figures, in this low-energy model, the relative displacement of Fermi surfaces of opposite spin results in a spin-split odd-parity-wave band structure. The band pairs with opposite out-of-plane spin polarization form a nodal crossing along the $k_y$. While the direction of the polarization of the states is out of the plane and protected by the spin symmetries of the model, we emphasize that the spin-polarization magnitude of the *p*-wave magnet is not protected, varying across the Brillouin zone. This *k*-dependent spin polarization gradient is essential for the finite NREE in the single-particle models of spin-split bands related by time reversal (see Supplementary Fig. 1 for more details).

To compute the non-equilibrium spin density accumulation $\delta\mathbf{S}$ due to an electric field, we use the Kubo linear response theory as $\delta S^i = \chi_S^{ij}E^j$, where $\mathbf{E}$ is the applied electric field. The spin-current response function ($\chi_S^{ij}$) has the following expression

$$\chi_S^{ij} = \frac{1}{2\pi}Re\sum_{\mathbf{k}\alpha\beta} S_{\alpha\beta}^i(\mathbf{k})v_{\beta\alpha}^j(\mathbf{k})\left[G_{\mathbf{k}a}^R G_{\mathbf{k}b}^A - G_{\mathbf{k}a}^R G_{\mathbf{k}b}^R\right]. \quad (2)$$

Here $G_{\mathbf{k}\alpha}^{R(A)} = 1/(\epsilon_F - \epsilon_{\mathbf{k}\alpha} \pm \frac{i\hbar}{2\tau_{\mathbf{k}\alpha}})$ is the retarded (advanced) Green's function at an energy $\epsilon_{\mathbf{k}\alpha}$, evaluated with respect to the Fermi energy $\epsilon_F$. $\tau_{\mathbf{k}\alpha}$ is the quasiparticle lifetime, taken here to be constant, $\sim \hbar/\Gamma$, where $\Gamma$ is the spectral broadening. We can separate the spin density into two parts depending on the contributions from the intra-band, $\delta\mathbf{s}_{\text{intra}}$ and inter-band, $\delta\mathbf{s}_{\text{inter}}$. Here, $S_{\alpha\beta}^i$ represents the $i^{th}$ component of the spin operator acting between states $\alpha$ and $\beta$. Similarly $v_{\beta\alpha}^j$, it represents the $j^{th}$ component of the velocity operator acting between $\beta$ and $\alpha$ states, respectively. The intra-band term with fermi surface contribution has the following expression[47,48],

$$\delta\mathbf{s}_{\text{intra}} = \frac{e\hbar}{2\Gamma} \int \frac{d^2k}{(2\pi)^2} \sum_\alpha \mathbf{S}_{\mathbf{k}\alpha} (\mathbf{E} \cdot \mathbf{v})_{\mathbf{k}\alpha} \delta(E_{\mathbf{k}\alpha} - E_F). \quad (3)$$

The $\delta\mathbf{s}_{\text{intra}}$ has the dominant contribution to charge spin conversion. The inter-band term can be expressed as,

$$\delta\mathbf{s}_{\text{inter}} = e\hbar \int \frac{d^2k}{(2\pi)^2} \sum_{\alpha\neq\beta} (f_{\mathbf{k}\alpha} - f_{\mathbf{k}\beta}) \text{Im}[\mathbf{S}_{\alpha\beta}(\mathbf{E} \cdot \mathbf{v})_{\beta\alpha}]$$
$$\times \frac{(E_{\mathbf{k}\alpha} - E_{\mathbf{k}\beta})^2 - \Gamma^2}{[(E_{\mathbf{k}\alpha} - E_{\mathbf{k}\beta})^2 + \Gamma^2]^2}. \quad (4)$$

Under time reversal symmetry (TRS) $S \to -S$ and $\mathbf{v} \to -\mathbf{v}$. Therefore, $\delta S_{\text{intra}}$ is allowed under TRS. However, the kernel of $\delta S_{\text{inter}}$ has very similar symmetry properties as the Berry curvature of the system, which vanishes if the system preserves TRS. For this reason, the 'intra' and 'inter' band components are often referred as $\mathcal{T}^{even}$ and $\mathcal{T}^{odd}$ part of the Edelstein response tensor. As $\mathcal{T}^{odd}$ is the odd Fermi surface property under time-reversal symmetry, it is only allowed for ferromagnets, altermagnets and some non-collinear AFMs breaking TRS[33,49] but prohibited in the $p$-wave candidates, as we also verify in our numerical calculations. For the identified $p$-wave material candidates in ref. 35 from the materials listed in MAGNDATA[50], we have included the symmetry-allowed tensorial form of the susceptibility in Table-I of Supplementary Information.

For our minimal model Hamiltonian, the $\chi_S^{zx}$ is the only finite component of the symmetry-constrained rank-2 tensor (Fig. 2d). The maximum value of the $\chi_S$ increases with $t_J$ due to the increasing anisotropy between the spin channels. The particle-hole symmetry evidenced in the response (Fig. 2d) is in agreement with the model band structure shown in Fig. 2b. In order to compare to other systems, we look at $\chi_S$ within (i) a non-coplanar AFM model with broken TRS with 3Q spin-texture[33] and (ii) a two-dimensional electron gas (2DEG) with relativistic Rashba SOC. For the former case, the non-equilibrium susceptibility, integrated over the unit cell, is reported to be $\chi_S^{3Q-AFM} \sim 0.5\hbar$ Å/V[33]. The non-equilibrium spin density for the Rashba 2DEG is given by $\delta S^R/eE = \alpha_R m_0/\hbar^2 \pi l^6$. We set $\Gamma = 0.01$ eV, $\alpha_R = 10^{-9}$ eVm as the typical Rashba strength expected in transition metal heterostructures[41,51], with $m_0$ being the free electron mass. This gives an effective Edelstein response $\chi_S^R \sim 0.4\hbar/V$ Å. Already at the simple model level, the strength of $\chi_S$ of the minimal $p$-wave model is substantially larger than that of the reported non-relativistic non-coplanar 3Q-AFM and relativistic Rashba 2DEG model.

## Tight Binding model of $p$-wave bi-kagome magnet

We next analyze the NREE in $p$-wave magnets within a multi-orbital two-dimensional Kagome lattice, as described by the Hamiltonian,

$$H = \sum_i \mathcal{J}_i \cdot \boldsymbol{\sigma} \, c_i^\dagger c_i + t_h \sum_{<ij>} c_i^\dagger c_j. \quad (5)$$

Here, $c^\dagger$ and $c$ are the fermionic creation and annihilation operators. $i, j$ are the site indices. The first term in Eq. (5) represents the interaction term where $\mathcal{J}_i$ as the local exchange parameter. $\boldsymbol{\sigma}$ is the vector containing three spin Pauli matrices. The kinetic energy of the Hamiltonian

is included in the second term, where $t_h$ is the isotropic nearest neighbor hopping strength. We choose the spin direction for six-inequivalent sites to be $\hat{\mathcal{J}}_1 = -\hat{\mathcal{J}}_4 = (\cos\theta_s \hat{x} + \sin\theta_s \hat{y})$, $\hat{\mathcal{J}}_2 = -\hat{\mathcal{J}}_5 = (-\cos\theta_s \hat{x} + \sin\theta_s \hat{y})$, $\hat{\mathcal{J}}_3 = -\hat{\mathcal{J}}_6 = -\hat{y}$ with $\theta_s = \frac{\pi}{6}$. The coplanar noncollinear magnetic ordering shown in Fig. 3a fulfils the spin symmetry criteria for the collinear $p$-wave spin-polarization band structure presented earlier. The neighboring red and blue triangular sub-units containing opposite $\Gamma_{4g}$ phases[52], are connected by $\mathcal{T}\mathbf{t}$ symmetry as shown by the black arrow. Although the 120° spin-order preserves the parity for a single Kagome unit cell, the $\mathcal{T}\mathbf{t}$ symmetric spin-order on the Kagome lattice breaks the inversion. Here, our focus is on materials whose non-collinear coplanar magnetic configuration is stabilized by means of higher-order bi-quadratic and four-spin ring exchange within a nearest-neighbor anti-ferromagnetic Heisenberg Kagome or triangular lattice, even in the absence of spin-orbit coupling[53]. However, we point out that there are also examples of non-collinear spin configurations stabilized by SOC, as observed in $CsCrF_4$[54,55]. As already emphasized before, due to the $[C_{2\perp}||\mathbf{t}]$ spin symmetry, the in-plane $S_x$ and $S_y$ components of the spins are identically zero for every momentum in the non-relativistic limit. The $\langle S_z \rangle$ polarized band structure along the high-symmetry $-M$-$\Gamma$-$M$ direction is shown in Fig. 3b. Here we choose $|\mathcal{J}| = |t_h| = 1.0$ eV throughout our model calculations.

Our calculations reveal a significant NREE for the Kagome TB model. We plot the $\chi_S$ response choosing the direction of the applied electric field along $x$ with $\Gamma = 0.01$ eV in Fig. 3c. The strength of the NREE susceptibility is of the same order as in the minimal $p$-wave model. We also show in Fig. 3d the expected $p$-wave anisotropy in the directional dependence of $\chi_S$, by rotating the $\mathbf{E}$ in the $x-y$ plane and plotting the $\chi_S$ for a fixed energy $E - E_F = -0.1$ eV in Fig. 3d. The black and white circles represent the positive and negative signs of the NREE susceptibility. The $\chi_S$ shows a nodal line for $\mathbf{E}||(-\frac{\sqrt{3}}{2}\hat{x} + \frac{1}{2}\hat{y})$ independent of the chemical potential. Hence, the angular measurement of the NREE provides a strong measurable signature for this NREE in these $p$-wave magnets.

We next explore the NREE dependence on the chirality of spins within the Kagome lattice, by adjusting the canting angle between the spins $\theta$ (inset of Fig. 3a, e). As a result, the $C_3$ symmetry of neighboring spins within individual triangles is disrupted, while the overall $[C_2||\mathbf{t}]$ connecting the red and blue triangles is preserved. The change in spin canting configuration changes the $\chi_S$ anisotropy pattern from that of the 120° order (see Supplementary Fig. 3 for more details). For $\theta_s = 90°$, $\chi_s$ vanishes since the spin-order becomes collinear, promoting spin-degenerate bands, consistent with the result from the full classification and delimitation of collinear spin arrangements on lattices by the spin symmetries[36]. In Fig. 3e we have plotted the NREE for two different energy values for $\theta_s$ in the range of $0° - 180°$. The increase in canting angle relative to the collinear arrangement ($\theta_s = 90°$) enhances the NREE and changes sign at $\theta_s = 90°$.

## Material candidate: CeNiAsO

We next compute the NREE for the realistic $p$-wave magnetic material candidate CeNiAsO[35]. The system shows a co-planar commensurate magnetic order with a moment $0.37\,\mu_B$ below Néel temperature, $T_N = 7.6K$[46]. From the spin symmetry analysis, the experimentally reported coplanar non-collinear magnetic order of CeNiAsO, as shown in Fig. 4a, b, implies the coplanar spin only group symmetry $\mathbf{r}_s = \{E, \bar{C}_{2z}\}$, where $\bar{C}_{2z}$ is 180° rotation around $z$-axis combined with spin-space inversion (time reversal). The nontrivial spin-space group $G^s$ in CeNiAsO contains the following symmetry elements[35,56,57]: $[E||E]$, $[E||\mathcal{M}_y \mathbf{t}_{\frac{b}{2}}]$, $[C_{2z}||\mathbf{t}_{\frac{a}{2}}]$, $[C_{2z}||\mathcal{M}_y \mathbf{t}_{(\frac{a}{2}+\frac{b}{2})}]$, $[C_{2x}||C_{2y}\mathbf{t}_{\frac{b}{2}}]$, $[C_{2y}||C_{2y}\mathbf{t}_{(\frac{a}{2}+\frac{b}{2})}]$, $[C_{2x}||P]$, $[C_{2y}||P\mathbf{t}_{\frac{a}{2}}]$, where $C$, and $\mathcal{M}$ denote rotation and mirror operations, and $\mathbf{t}_i$ is the translation along $i$ of the lattice vector. The symmetry operation $[C_{2z}||\mathbf{t}_{\frac{a}{2}}]$ (see Fig. 4a) in combination with broken inversion symmetry fulfils the above symmetry conditions for the odd-parity-wave

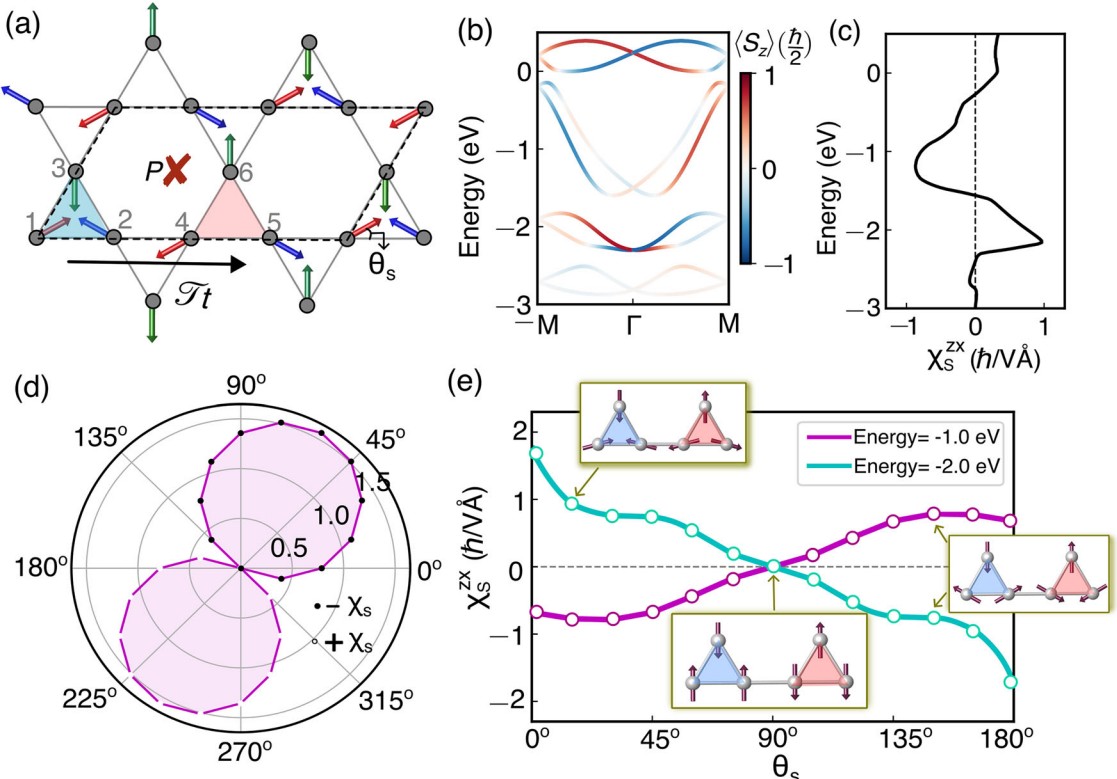

**Fig. 3 | Dependence of Edelstein response on electric field direction and spin orientation for $\mathcal{T}t$ symmetric bi-kagome lattice. a** Direct space magnetic order of the Kagome lattice with twice propagation along the $x$-axis. The magnetic order breaks the inversion and exhibits combined TRS and lattice translation symmetry, $\mathcal{T}t$ marked by the black arrow, which connects the spins in the blue and red shaded triangles. The coplanar spins in each magnetic triangle observe 120° spin alignment. The spin canting angle of the red spin, $\theta_s$, is also marked. **b** The out-of-plane spin-projected band dispersion along $-M$-$\Gamma$-$M$ high-symmetry direction. **c** The non-equilibrium intra-band susceptibility density, $\chi_S^{zx}$, for electric field $\mathbf{E} = E\hat{x}$. **d** The angular distribution of $\chi_S^{zx}$ (in units of $\hbar/V$ Å) for energy$-$-0.1 eV w.r.t the electric field direction. The white and black circles represent the positive and negative sign of $\chi_S$ (**e**). The magnitude of $\chi_S^{zx}$ gradually increases with the spin canting angle $\theta_s$, relative to the collinear arrangement at $\theta_s = 90°$.

magnetic state. Additionally, the symmetry $[C_{2y}||C_{2y}]$ enforces a single spin-unpolarized line the $k_x = k_z = 0$ in the $k_z = 0$ plane, implying $E(k_x, k_y, k_z, \sigma_z) = E(-k_x, k_y, -k_z, -\sigma_z)$. The fixed energy contour at $E - E_F = -0.1$ eV and at $+0.12$ eV within $k_x - k_y$ plane (Fig. 4c, d) shows a nodal line along $k_x = k_z = 0$ line. We plot the band dispersion of CeNiAsO in Fig. 4e. The bands have opposite out-of-plane spin polarization $S_z$ for opposite momentum with a linear crossing at the $\Gamma$ point. The anti-symmetric $S_z$ projection is a direct consequence of $\mathcal{T}$ being a point group symmetry operation for CeNiAsO, which also restricts the responses to be time-reversal-even even in the presence of SOC. We note that there can be some co-planar magnetic ordered systems that do not have TRS as a point group symmetry element but have extra spin-symmetries that lead to the in-plane spin-component in the non-relativistic band structure to become zero, as in e.g. ref. 31. This leads to an antisymmetric out-of-plane spin splitting in the non-relativistic band structure. However, in the presence of SOC, these systems that lack TRS as point group symmetry no longer have an antisymmetric spin-splitting band structure and, in fact, allow for time-reversal-odd responses as well.

We compute all the components of the NREE susceptibility for metallic CeNiAsO in the limit of zero SOC, shown by solid lines in Fig. 4f. Our spin symmetry analysis (see Supplementary Table 1 for details) yields that only the $\chi_S^{zx}$ and $\chi_S^{zz}$ components survive, which is consistent with our numerical results. Given the negligible van-der-Waals interaction between the Ce layers stacked along the $z$-axis within the bulk geometry, we find a small $v_z$ component of the velocity. Therefore, the obtained $\chi_S^{zz}$ value is substantially lower than that of $\chi_S^{zx}$.

Figure 4c, d represent the isoenergy contours at energies where the NREE susceptibility $\chi_S^{zx}$ reaches the maximum and minimum values, as shown with dotted lines in Fig. 4f.

Of course, along with non-relativistic exchange-driven effects, SOC can also have an impact (see Supplementary Fig. 4) due to the fact that Ce is a rare-earth element. Within the relevant magnetic point group analysis, by its own construction, the spin group symmetry $[C_{2\perp}||\mathbf{t}]$ is broken by spin-orbit coupling and, therefore, no longer enforces the polarization direction to be out of the plane. The induced finite in-plane spin components hence emerge solely due to the SOC effect. In Fig. 4f, we show the change of the NREE susceptibility non-zero components in the presence of SOC with dashed curves. The susceptibility components emerging solely from the relativistic origin are plotted in Fig. 4g. The details of relativistic band dispersion and the corresponding tensorial form of the non-equilibrium spin accumulation susceptibility are included in section 5 of the Supplementary Information. While the NREE dominates prominently over a large range of energies, the contributions of the SOC susceptibilities of in-plane polarization can be exploited to calibrate the relevance of this contribution in general. Comparing the relative magnitude of the in-plane and out-of-plane EE susceptibilities should be a good experimental test of the relevance of the NREE.

Next, we compare the magnitude of NREE susceptibility of $\mathcal{T}t$ symmetric $p$-wave magnets with recently reported non-relativistic spin accumulation within non-collinear magnets with broken TRS, i.e. non $p$-wave magnets. The NREE reported for non-centrosymmetric LuFeO$_3$ is $0.5\,\hbar$ Å$/V$ calculated within the unit cell of volume 360.61 Å$^3$[33]. The

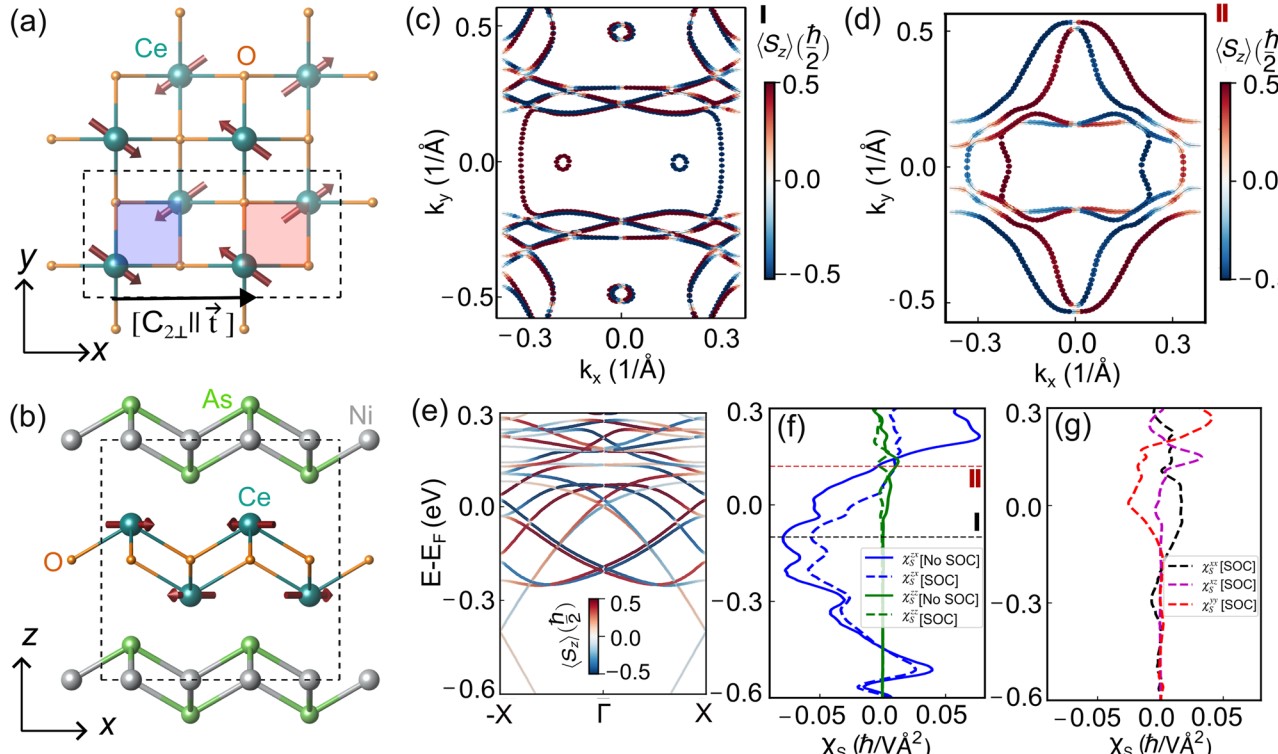

**Fig. 4 | Material realization of nodal $p$-wave character in CeNiAsO. a** Top and **b** side view of the unit cell of CeNiAsO crystal and the co-planar magnetic order with $\mathcal{T}t$ symmetry. The out-of-plane spin projected constant energy isosurface at $E \sim -0.1$ eV **c** and $E \sim 0.12$ eV **d** in the $k_z = 0$ momentum plane. **e** Nonrelativistic odd-parity wave spin splitting of energy bands plotted along the $-X$-$\Gamma$-$X$ path. The colormap represents the out-of-plane $\langle S_z \rangle$ component. **f** The symmetry-allowed components of non-equilibrium intra-band susceptibility density, $\chi_S^{zx}$ and $\chi_S^{zz}$. The

modifications of the specific components in the presence of SOC are shown in dashed curves. The isoenergy cuts of (**c**) and (**d**) are at the energy where the $\chi_S^{zx}$ component of the NREE susceptibility acquires maximum and minimum values, as indicated by dotted lines in **f**. **g** Susceptibility components arise due to the sole relativistic effect. SOC leads to finite $\chi_S^{xx}$, $\chi_S^{xz}$ and $\chi_S^{yy}$ which are not allowed within the NREE susceptibility tensor.

effective $\chi_S^{zx}$ integrated over the unit cell of volume $267.50\,\text{Å}^3$ of CeNiAsO gives NREE $\sim 13\,\hbar\,\text{Å}/V$, which is 25 times higher in magnitude than that of NREE reported for LuFeO$_3$. Here, to make a systematic comparison of the charge-to-spin conversion for $p$-wave CeNiAsO and co-planar LuFeO$_3$, we choose the same value of $\Gamma$. Since scattering times can vary with temperature and disorder, to more directly compare the systems we examine a $\Gamma$-independent parameter, $\chi_S/\sigma_D$, where $\sigma_D$ represents the Ohmic conductivity, as detailed in Supplementary Fig. 5. We find that around the Fermi energy, even the minimum value of $\chi_S/\sigma_D$ for CeNiAsO around the Fermi energy exceeds by more than 10 times that of the maximum of LuFeO$_3$. This highlights that CeNiAsO, with its p-wave characteristics, stands out as a promising platform for studying giant anisotropic NREE, making it a compelling candidate for future investigations in spintronic applications.

## Discussion

We predict here a highly efficient NREE in nodal $p$-wave magnets. This type of coplanar noncollinear magnetic order preserves TRS in momentum space (through the $\mathcal{T}t$ symmetry) while exhibiting parity-polarized band structure. Our nodal $p$-wave NREE inherits the distinctive features of the unconventional $p$-wave magnets and is thus distinct from the conventional relativistic Rashba-Edelstein effect in three aspects: First, the spin symmetry $[C_{2\perp}||\mathbf{t}]$ of $p$-wave magnets forces the polarization direction in the band structure to be perpendicular to the plane of spin coplanarity. This can give rise to out-of-plane nonequilibrium spin accumulation, a feature not permitted in the usual relativistic Rashba-Edelstein effect and in other NREEs in noncoplanar magnets. Second, the unique anisotropic momentum-dependent splitting in $p$-wave magnets gives them an anisotropic

nodal EE signature, which is absent in the relativistic but also other magnetic system counterparts. Third, the non-equilibrium spin accumulation within $p$-wave magnets does not originate from the SOC effect but rather from the exchange interaction of the non-collinear magnetic order. In turn, NREE in $p$-wave magnets can lead to unprecedented high efficiency in spin-to-charge conversion. The inevitable presence of SOC can lead to contributions to the EE within these materials, particularly in the presence of heavy elements, and its contribution to the spin-charge conversion efficiency can be estimated by the measured in-plane vs. out-of-plane susceptibilities. However, this contribution is expected to be significantly less pronounced in material candidates with lighter elements. We predict here that the charge-to-spin conversion in the metallic material candidate CeNiAsO, showing coplanar noncollinear magnetic order, is in a large energy window dominated by the NREE.

## Methods

We have used density functional theory (DFT) in the plane wave basis set. We used the Perdew-Burke-Ernzerhof (PBE)[58] implementation of the generalized gradient approximation (GGA) for the exchange-correlation. This was combined with the projector augmented wave potentials[59,60] as implemented in the Vienna ab initio simulation package (VASP)[61,62]. The kinetic energy cutoff of the plane wave basis for the DFT calculations was chosen to be 460 eV. A $\Gamma$-centered $4 \times 8 \times 4$ $k$-point grids are used to perform the momentum-space calculations for the Brillouin zone (BZ) integration. We have constrained the magnetic moments without incorporating SOC and switching off the crystal symmetry to capture the sole non-relativistic effect. A penalty contribution to the total energy is considered in these

calculations to constrain the moments. The penalty energy fixes the local moment into a specific direction[63]

$$E = E_0 + \sum \gamma \left[ \mathbf{M}_i - \mathbf{M}_i^0 \left( \mathbf{M}_i^0 \cdot \mathbf{M}_i \right) \right]^2, \quad (6)$$

where $E_0$ is the DFT energy without any constraint, and the second term represents the penalty energy contribution due to the non-collinear direction constraint. $\mathbf{M}_i^0$ and $\mathbf{M}_i$ represent the unit vector along the desired direction of magnetic moment and the integrated magnetic moment inside the Wigner-Seitz cell at site $i$, respectively. The sizes of the Wigner size radii are to be chosen as 1.98 Å, 1.21 Å, 1.09 Å and 0.24 Å for Ce, Ni, As and O atoms, respectively. The choice of $\gamma$ controls the penalty energy contribution. In our calculations for CeNiAsO, we set $\gamma = 6 \, \text{eV}/\mu_B^2$ to get a negligible penalty contribution $1 \times 10^{-8}$ eV. The variation of magnetic moment at Ce sites for different choices of Hubbard $U$ are included in section-7 of Supplementary Information. We construct the tight-binding model Hamiltonian of CeNiAsO by using atom-centered Wannier functions within the VASP2WANNIER90[64] codes. Utilizing the obtained tight-binding model, we calculate the Edelstein response tensor.

## Data availability
The data that support the findings of this study are all available from the main text and supplementary information.

## Code availability
The codes used to calculate material-specific electronic structure are described in detail in the Methods section. The in-house code for the models and to calculate Edelstein response, which supports the findings of this study, is available from the corresponding authors upon reasonable request.

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

## Acknowledgments

JS, AC, ABH, and LS acknowledge funding by the Deutsche Forschungsgemeinschaft (DFG, German Research Foundation) - TRR 173 - 268565370 (project A03) and TRR 288 - 422213477 (project A09 and B05), and the Alexander von Humboldt Foundation. TJ acknowledges support by the Ministry of Education of the Czech Republic, CZ.02.01.01/00/22008/0004594 and ERC Advanced Grant no. 101095925. LS acknowledges support from the ERC Starting Grant No. 101165122. We acknowledge the high-performance computational facility of the supercomputer 'Mogon' at Johannes Gutenberg Universität Mainz, Germany. The authors acknowledge fruitful discussions with Gerrit Bauer, Rafael González Hernández, and Nayra A. Alvarez Pari.

## Author contributions

J.S., L.S., T.J. and A.C. conceived and designed the project. A.C. conducted the theoretical modeling and first-principles calculations. A.C. and A.B.H. constructed the minimal model Hamiltonian. A.C. and R.J.U. did the symmetry analysis of the response tensors. A.C. and J.S. co-wrote the manuscript with feedback from all the coauthors. All the authors discussed the results and reviewed the manuscript.

## Funding

## Competing interests

The authors declare no competing interests.
