## [Peer Review File · Nature Communications]

Highly Efficient Non-relativistic Edelstein effect in nodal p-wave magnets

Corresponding Author: Dr Atasi Chakraborty

Version 0:

Reviewer comments:

Reviewer #2

(Remarks to the Author)

By using the effective model, tight-binding model, and DFT calculations, combined with the Kubo formula, the authors study the Edelstein effect in the so-called non-relativistic p-wave magnets. The authors argue that a large response of the Edelstein effect can be obtained in such p-wave magnets. In my view, the idea of this work is interesting, the results shown in this work seem to be reliable and reasonable. However some parts of this work are hard to follow, and some necessary explanations for these formulas are lacking. To conclude, I do not think the current version of this work is suitable for publication in Nature Communications. The following questions should be answered properly.

1, For Equations (2), (3.) and (4) on Page 5, every symbol should be explained. Please do not let your readers guess what these symbols mean. In the No. 100 line on Page 5, what is "Ref. Hellenes2023"?

2, For the tight-binding model on the 2D bi-Kagome lattice, the authors propose a spin texture shown in Figure 3, which keeps the required symmetry for p-wave magnets. While, in my view, once such kind of spin-texture is stabilized in the bi-kagome lattice, the spin-orbit coupling is required. Without the spin-orbit coupling, which kind of interaction could stabilize this spin texture?

3, As the authors mentioned in the paper, the Edelstein effect depends on the scattering time of carriers around the Fermi level, which is determined by the scattering mechanisms and the electronic properties of specific material candidates. It is meaningless to mention a specific value to compare with NREE in other magnetic materials.

4, On page 6, the authors use Γ as 0.1 eV when estimating effective Edelstein response for the typical Rashba materials. While, when the authors estimate NREE susceptibility for the Kagome model, they choose Γ as 0.01 eV. WHY? How large is the value of Γ in the calculation for CeNiAsO?

5, For the DFT calculation in CeNiAsO, how did the authors deal with the f-electron? Does the electronic structure of CeNiAsO depend on on-site U of f-electron? Maybe the authors should compare the results with those from DMFT calculations.

Reviewer #3

(Remarks to the Author)

In this manuscript, Atasi Chakraborty et al. present a study on the non-relativistic Edelstein effect, NREE, in p-wave magnets, in other words, the Edelstein effect arising from specific spin arrangement rather than spin-orbit coupling (SOC).

Since it has been known that non-collinear magnetism mimics SOC to some extent and can support some "SOC-like" phenomena without SOC, such as spin splitting, electric-dipole spin resonance, spin Hall effect, and so on, it is natural to try to figure out if the Edelstein effect which was previously considered closely related to SOC (the earliest research of which was studied in the Rashba system) can also be realized without SOC. Several related works have already been published recently, such as the study on NREE in non-collinear magnets, which was published in Nature Communications by Rafael González-Hernández et al. in September 2024.

Below, I have enumerated several questions and comments that I would like the authors to address before I make a final decision on this manuscript.

1. In the second paragraph, the authors try to compare their NREE in p-wave magnets with NREE in other systems and traditional relativistic EE. The key point they keep emphasizing is the time-reversal symmetry (TRS). So here are several questions:

(1) To me, TRS is not directly related to the Edelstein effect, which only requires the broken of inversion symmetry. Then why do we need to care about the broken or preservation of TRS in different systems?

(2) What is explicitly the TRS and TRS breaking in momentum space?

(3) The TRS they are talking about in p-wave magnets in momentum space is actually not the real TRS, but an alternative one from Tt (t is translation) in the real space. More than just Tt , a system can break TRS but preserve TU , TUt (U is the proper spin rotation) in the real space but still have alternative TRS in the momentum space. So from this point of view, the papers they cite to claim that other NREE systems have no TRS are actually do have alternative TRS in the momentum space, which is just the same as their work here.

2. The authors claim that the response, or more specifically, susceptibility, of their NREE in p-wave magnet is several times larger than in other systems, which is the main advantage of their NREE over other systems. However, In the work published by Nayra A. Álvarez Pari et al. arXiv:2412.10984, they calculated a one times larger susceptibility in the non-collinear EuIn_2As_2 system than in CeNiAsO . What is the reason for that?

In conclusion, while the concept of non-relativistic Edelstein effect is not entirely new, the p-wave magnet has not yet been used as a research system for NREE. But apart from that, this work seems to lack enough novelty to be published in Nat. Commun. Moreover, the advantage they emphasize of the large susceptibility \ response in CeNiAsO might be larger than normal EE, but is indeed not that surprisingly large in the NREE region.

Version 1:

Reviewer comments:

Reviewer #2

(Remarks to the Author)

The authors answered my questions properly. I suggest the publication of this paper in Nature Comm.

Reviewer #3

(Remarks to the Author)

The authors have properly addressed most of my questions and modified the manuscript accordingly. However, there is still several points that need to be addressed before I can recommend this work for publication:

1. While it is acknowledged that Tt (time-reversal T combined with a translation t operation) possesses certain advantages, this does not imply that TU (U is a proper spin rotation) is inherently deficient. Consider a scenario where the effective time reversal operation of the system is TU rather than Tt . When the introduced spin-orbit coupling (SOC) is sufficiently small, the presence of spin splitting can still be sustained, thereby permitting the occurrence of NREE. Conversely, when the SOC becomes sufficiently large, the existence of in-plane spin components—even in the case of a p-wave magnet (pM)—results in a reduction of spin accumulation, that is, the response associated with NREE, due to these in-plane components. It seems to me that there is no remarkable difference between TU (also TUt) and Tt if SOC is included. Is it true? Moreover, if the analysis is confined to spin accumulation along the z-direction, $M_z T$ can produce an effect equivalent to that of Tt , correct?

2. I also want to address an issue concerning p-wave magnets. The author asserted that a pM must possess a nodal plane; however, there should exist nodeless pMs, corresponding to the chiral p-wave superconductor in superconductivity. This is exemplified by the beta phase pM, as noted in arXiv:2504.14577v1.

3. I am also curious about the rigorous definition of P-wave magnet. Does it need to forbid all the even-order polynomials of the $k \cdot p$ Hamiltonian? For example if the high-symmetry point has D_2 magnetic group symmetry, then even-order polynomials are also forbidden. Then will it be a p-wave magnets even considering SOC? I would highly recommend the authors to provide a thorough definition and symmetry requirement of p-wave magnets.

Version 2:

Reviewer comments:

Reviewer #3

(Remarks to the Author)

Recommend publication.

We thank the reviewers for carefully reviewing our manuscript. Their feedback has helped us to improve the readability of our manuscript and to make some important points clearer. Below, we address the reviewers comments. The reviewers comments/questions are in **blue colored font**, and our responses are in black colored font. All the changes in the revised manuscript are marked in **red colored font**.

Reviewer #2:

By using the effective model, tight-binding model, and DFT calculations, combined with the Kubo formula, the authors study the Edelstein effect in the so-called non-relativistic p-wave magnets. The authors argue that a large response of the Edelstein effect can be obtained in such p-wave magnets. In my view, the idea of this work is interesting, the results shown in this work seem to be reliable and reasonable. However some parts of this work are hard to follow, and some necessary explanations for these formulas are lacking. To conclude, I do not think the current version of this work is suitable for publication in Nature Communications. The following questions should be answered properly.

Reply: We thank the referee for the careful reading of the manuscript and for appreciating our work. Below, we address all the questions and suggestions raised by the referee, and indicate how we have modified the revised version of the manuscript accordingly.

1, For Equations (2), (3), and (4) on Page 5, every symbol should be explained. Please do not let your readers guess what these symbols mean. In the No. 100 line on Page 5, what is “Ref. Hellenes2023”?

Reply: We fully agree with the referee that every term in the equations should be defined and that there should be no ambiguity in the meaning of any of the terms.

Action taken: Following the reviewer’s suggestion, we have explicitly defined all the symbols used in equations (2), (3) and (4). We have also corrected the typographical error in the citation, replacing “Ref. Hellenes2023” with Ref. [35]. The text below equations (2) now reads:

“Here, $G_{\mathbf{k}\alpha}^{R(A)} = 1/(\epsilon_F - \epsilon_{\mathbf{k}\alpha} \pm \frac{i\hbar}{2\tau_{\mathbf{k}\alpha}})$ is the retarded (advanced) Green’s function at an energy $\epsilon_{\mathbf{k}\alpha}$, evaluated w.r.t the Fermi energy ϵ_F . $\tau_{\mathbf{k}\alpha}$ is the quasiparticle lifetime, taken here to be constant, \hbar/Γ , where Γ is the spectral broadening. We can separate the spin density into two parts depending on the contributions from the intra-band, $\delta\mathbf{s}_{\text{intra}}$ and inter-band, $\delta\mathbf{s}_{\text{inter}}$. Here, $S_{\alpha\beta}^i$ represents the i^{th} component of the spin operator acting between states α and β . Similarly $v_{\beta\alpha}^j$ represents the j^{th} component of the velocity operator acting between β and α states respectively.”

2, For the tight-binding model on the 2D bi-Kagome lattice, the authors propose a spin texture shown in Figure 3, which keeps the required symmetry for p-wave magnets. While, in my view, once such kind of

spin-texture is stabilized in the bi-kagome lattice, the spin-orbit coupling is required. Without the spin-orbit coupling, which kind of interaction could stabilize this spin texture?

Reply: Indeed, the referee is correct that spin-orbit coupling (SOC) can stabilize the $\mathcal{T}\vec{t}$ symmetric 120° magnetic order. Here \mathcal{T} is time reversal symmetry and \vec{t} is a translation. A good example of this case is the p-wave candidate CsCrF_4 containing local kagome networks, where similar magnetic ordering emerges due to the interplay of single-ion anisotropy, Dzyaloshinskii-Moriya interaction, and Heisenberg exchange –each essential in stabilizing this quasi- 120° structure [npj Quantum Materials 4, 14 (2019), Phys. Rev. B 91, 224403 (2015)].

However, it is also possible to stabilize this non-collinear $\mathcal{T}\vec{t}$ symmetric 120° magnetic order in the absence of SOC within a nearest neighbor anti-ferromagnetic Heisenberg Kagome or triangular lattice by means of bi-quadratic exchange interactions and four-spin ring exchange [Nat. Commun. 11, 511 (2020)]. Our focus is on such materials, which would yield the strongest non-relativistic effects.

Action taken: We have included the following statements in page 7 of the revised manuscript in the context of the stabilization of the chosen non-collinear spin configuration within bi-kagome network: “*Here our focus is on materials whose non-collinear coplanar magnetic configuration is stabilized by means of higher-order bi-quadratic and four-spin ring exchange within a nearest neighbor anti-ferromagnetic Heisenberg Kagome or triangular lattice, even in the absence of spin-orbit coupling [52]. However, we point out that there are also examples of non-collinear spin configurations stabilized by SOC, as observed in CsCrF_4 [52, 53].*”

3, As the authors mentioned in the paper, the Edelstein effect depends on the scattering time of carriers around the Fermi level, which is determined by the scattering mechanisms and the electronic properties of specific material candidates. It is meaningless to mention a specific value to compare with NREE in other magnetic materials.

Reply: We thank the referee for this insightful comment. Since the non-equilibrium spin accumulation scales with the scattering time (τ), its exact quantitative value can vary somewhat depending on the system. A simple rule of thumb for simple metals gives a scattering time that can be roughly estimated using the following formula

$$\tau = \left(\frac{0.22}{\rho}\right) \left(\frac{r_s}{a_0}\right)^3 \times 10^{-14} \text{ sec.} \quad (1)$$

Here, ρ , r_s and a_0 represent resistivity (in micro-ohm cm), the radius of a sphere with a volume equal to the volume per conduction electron and Bohr radius, respectively. At room temperature, τ typically falls within the range of 10^{-14} sec to 10^{-15} sec.

To ensure a more direct comparison between different systems, we have instead calculated the response for the same Γ ($\equiv \hbar/\tau$) value, i.e. assuming similar conductivities/disorder broadening. This compares the ratio

between efficiencies of spin-conversion of the two systems up to their proportionality on their dissipation.

Our findings indicate that the order of magnitude of the NREE response of p-wave systems is significantly larger (1-2 orders in magnitude) than any previous reported value, suggesting that such a large difference is unlikely to arise from the typically smaller variations in system-dependent scattering time. For this study, we used the $\Gamma = 0.01$ eV, following the widely chosen parameter value in literature for calculating spin-charge conversion (Phys. Rev. Lett. **113**, 157201 (2014), Nat. Commun. **15**, 7663 (2024), arXiv:2410.17993 (2024) etc.).

To address this issue raised by the referee, we look at the ratio of the NREE susceptibility (χ) relative to the longitudinal Ohmic conductivity (σ_D) for the two systems, which is τ independent. We compare our p-wave candidate CeNiAsO with the LuFeO₃ material reported Nat. Commun. **15**, 7663 (2024). In the top panel of the figure, we have plotted the Ohmic conductivity for CeNiAsO (RFig. 1a) and LuFeO₃ (RFig. 1b) around the Fermi energy (E_F) by using the following expression,

$$\sigma_{ij} = \frac{2\pi}{\Gamma} \left(\frac{e^2}{h} \right) \int [d\mathbf{k}] \sum_n \frac{\delta E_n}{\delta k_i} \frac{\delta E_n}{\delta k_j} \left(-\frac{\delta f_0}{\delta E} \right)_{E=E_n}. \quad (2)$$

Here, E_n is the energy for the n^{th} band and f_0 is the Fermi distribution function. We set $\Gamma = 0.01$ eV, an identical value that has been used to calculate the Edelstein responses. Interestingly, we see LuFeO₃ shows isotropic in-plane Drude conductivity, whereas, in CeNiAsO, the unconventional p-wave magnetic phase spontaneously breaks the crystal symmetry, resulting in a large resistive anisotropic longitudinal response. In panels RFig. 1(c) and RFig. 1(d), we have plotted the ratio of the highest component of \mathcal{T} -even Edelstein response with σ_D , to make it a scattering-time independent parameter. We have also included the zoomed-in view in the inset in the energy region where the σ_D is highest and expected to show minimum χ/σ_D values for CeNiAsO. We find that even the minimum value of the ratio for p-wave CeNiAsO surpasses by more than an order of magnitude the values calculated for LuFeO₃.

Action taken: We have included the following discussions at the end of section-1.3 (page 9) to highlight this scattering time aspect “*Here, to make a systematic comparison of the charge-to-spin conversion for p-wave CeNiAsO and co-planar LuFeO₃, we choose the same value of Γ . Since scattering times can vary with temperature and disorder, to more directly compare the systems we examine a Γ -independent parameter, χ_S/σ_D , where σ_D represents the Ohmic conductivity, as detailed in Section 6 of the Supplementary Information. We find that around the Fermi energy, even the minimum value of χ_S/σ_D for CeNiAsO around the Fermi energy exceeds by more than 10 times that of the maximum of LuFeO₃. This highlights that CeNiAsO, with its p-wave characteristics, stands out as a promising platform for studying giant anisotropic NREE, making it a compelling candidate for future investigations in spintronic applications.*”

RFig. 1. Longitudinal conductivities calculated along the x and y axis for (a) CeNiAsO and (b) LuFeO₃. The scattering time independent parameter (χ/σ_D) for (c) CeNiAsO and (d) LuFeO₃. Here, we have chosen the highest global Edelstein response tensor component to calculate the ratio, which is χ_{zx} and χ_{xy} for CeNiAsO and LuFeO₃, respectively.

We have also included the aforementioned comparison with respective plots of σ_D and χ/σ_D for p -wave CeNiAsO and coplanar LuFeO₃ in section 6, namely “*Scattering time independent comparison of response for CeNiAsO and LuFeO₃*”, of the SI in the revised manuscript.

4, On page 6, the authors use Γ as 0.1 eV when estimating effective Edelstein response for the typical Rashba materials. While, when the authors estimate NREE susceptibility for the Kagome model, they choose Γ as 0.01 eV. WHY? How large is the value of Γ in the calculation for CeNiAsO?

Reply: We thank the reviewer for pointing this out. We have now reported the effective Edelstein response for Rashba materials for 0.01 eV, aligning with the values chosen for the minimal model, the bi-kagome magnet and CeNiAsO. We chose $\Gamma = 0.01$ eV for CeNiAsO, the same value was used in LuFeO₃ (*Nat. Commun.* **15**, 7663 (2024)) with which we compared the NREE.

Action taken: We have included the sentence “*This gives an effective Edelstein response $\chi_S^R \sim 0.4\hbar/V\text{\AA}$.*” to report the Edelstein response of Rashba systems with the same Γ value as chosen for others. We have explicitly mentioned the Γ value i.e. “ $\Gamma = 0.01$ eV” for the models and the material candidates in the revised manuscript.

5, For the DFT calculation in CeNiAsO, how did the authors deal with the f-electron? Does the electronic structure of CeNiAsO depend on on-site U of f-electron? Maybe the authors should compare the results with those from DMFT calculations.

Reply: As a lanthanide element, cerium typically exhibits strong electron correlations at its atomic sites, which are essential for maintaining localized f -orbitals. The typical choice of Hubbard U for Ce atoms lie within the range of 4.3-6.7 eV [*J. Chem Phys* **140**, 084101 (2014)]. In our calculations, we used $\lambda = 6$ eV/ μ_B^2 , constraining non-collinear magnetic moments for different choices of Hubbard U at Ce sites within the GGA+ U scheme. In RFig. 2a, we have plotted the orbital projected band-dispersion of the Ce- f states along high-symmetry paths with $U = 0$ eV. We find that the calculation without Hubbard interaction can successfully capture the strongly localized nature of the Ce- f bands near the Fermi energy. Interestingly, the magnetic moment at the Ce atom i.e. $(\pm 0.35, \pm 0.26, 0) \mu_B$ within $U = 0$ eV calculations agrees well with the experimentally obtained effective moment value, $\mu_{eff} = (\pm 0.3, \pm 0.22, 0) \mu_B$ [*Phys. Rev. Lett.* **122** 197203 (2019)], below 7.6 K. The contribution of localized Ce- f states is closer to the conduction band edge (see RFig. 2a) as similar to the other high-temperature phases of CeNiAsO reported in the literature [*J. Phys.: Cond. Mat.* **23**, 175701 (2011)]

Our further theoretical calculations suggest, as per the expectation, that the increase of Hubbard U at the magnetic site enhances the magnetic moment at Ce sites, as shown in RFig. 2b. However, the increased moment for finite values of Hubbard U , the theoretically obtained moment, vastly deviates from the experimental effective moment (marked as a red star). Therefore, we have carried out our calculations of the electronic structure and responses with $U = 0$ eV.

RFig. 2. (a) The orbital decomposed band dispersion of Ce- f states (black) on the total energy dispersion. (b) Variation of magnetic moment at Ce site for different values of Hubbard U . The experimentally observed moment in μ -SR experiment is shown in *Phys. Rev. Lett.* **122**, 197203 (2019) is marked with a red star.

Action taken: We have included the discussion of Hubbard U in the new section-7, namely “*Effect of electronic correlation*”, in the SI (page 11) of the revised manuscript.

Reviewer #3:

In this manuscript, Atasi Chakraborty et al. present a study on the non-relativistic Edelstein effect, NREE, in p-wave magnets, in other words, the Edelstein effect arising from specific spin arrangement rather than spin-orbit coupling (SOC).

Since it has been known that non-collinear magnetism mimics SOC to some extent and can support some “SOC-like” phenomena without SOC, such as spin splitting, electric-dipole spin resonance, spin Hall effect, and so on, it is natural to try to figure out if the Edelstein effect which was previously considered closely related to SOC (the earliest research of which was studied in the Rashba system) can also be realized without SOC. Several related works have already been published recently, such as the study on NREE in non-collinear magnets, which was published in Nature Communications by Rafael González-Hernández et al. in September 2024.

Reply: We thank the reviewer for her/his careful review. Below we address point by point all the questions and concerns raised by the reviewer.

While recent reports on non-collinear magnets indicate the possibility of spin-charge conversion effects, there is not a direct comparison to the non-magnetic systems with broken inversion symmetry, e.g. Rashba 2D systems, where the effect was first observed. The systems considered in recent reports all have broken time reversal symmetry band structures - where for example effects as the anomalous Hall effect have been observed. The fundamental difference is that the p-wave magnets exhibit a time reversal symmetry element in the point group (which translates into a time reversal symmetric band structure with spin-splittings) in direct analogy to non-magnetic spin-orbit coupled systems.

Not only is there an expectation of a much larger spin-charge conversion effect in the p-wave magnetic candidates, but their phenomenology is strikingly unique. There exist key fundamental differences between p-wave magnets and non-collinear magnets with broken time reversal symmetry in terms of characteristics and responses, which we list below:

1. In the recent reports of non-collinear magnets (including *Nat. Commun.* **15**, 7663 (2024) as mentioned by the reviewer) featuring NREE, the crystal structure itself breaks inversion. However, in the *p*-wave magnet considered in our work, the broken inversion is exchange-driven and promoted solely by the magnetic order within a centrosymmetric crystal.

2. In p -wave magnets, the NREE shows a strong anisotropic response with an out-of-plane polarized spin density resulting from the spin symmetries, as we see in our model calculations and also for realistic material example CeNiAsO. This out-of-plane response is the unique property of the p -wave magnets, as the relativistic Rashba systems can only offer in-plane spin accumulation. The NREE of ref. 33 and 34, as well as other reports, show that there is no preferred uni-directionality and that all finite response components can occur depending on the systems. This directional averaging leads naturally to a much lower efficiency of the spin-charge conversion in these systems.
3. As already mentioned, for the p -wave magnets, the point group symmetry operations include the time reversal (\mathcal{T}) as an independent symmetry operation, which makes the direct analogy with the SOC non-magnetic systems, but now realized in a purely non-relativistic limit. We again explained this point in detail in answer to the 3rd question from a detailed symmetry analysis.

Additionally, the realistic examples discussed in Nat. Commun. 15, 7663 (2024) (as mentioned by the reviewer) the \mathcal{T} -even response tensor is finite only in the presence of SOC as mentioned in the sentences (page-6 first paragraph): “With-out spin-orbit coupling for the global effect, only the T-odd component is allowed (Fig. 4b). With spin-orbit coupling, even the global T-even component appears (Fig. 4a)” Whereas, for p -wave CeNiAsO, \mathcal{T} -even non-equilibrium spin accumulation arises even in the absence of SOC, as shown in Fig. 4f of the main manuscript.

Action taken: To make these key differences clear, we have included the above-mentioned comparison of electronic structure and responses of unconventional p -wave systems with non-collinear magnets featuring NREE in the ‘Introduction’ section of the main manuscript. The end of the second paragraph in the introduction emphasizing the uniqueness of our findings with respect to the recent reports of NREE now reads: *“The EE is, in principle, allowed if the system breaks inversion symmetry irrespective of the origin of the response, be it SOC or non-collinear order. The conventional relativistic Rashba-Edelstein effect in TRS systems originates from SOC induced anti-symmetric spin textures in momentum space. The recently predicted p -wave magnets (Fig. 1c) have TRS as a point group operation, thus exhibiting TRS in momentum space akin to Rashba systems. However, unlike Rashba spin splitting, which, is isotropic, our p -wave spin splitting is strongly anisotropic and momentum dependent [35]. This can give rise to out-of-plane non-equilibrium spin accumulation, a feature not permitted in the usual relativistic Rashba-Edelstein effect. This p -wave magnet NREE contrasts with recent reports [33, 34] that obtain a NREE in non-collinear magnetic order in non-centrosymmetric crystals and which do not have TRS as a point group symmetry. In p -wave magnets, the inversion symmetry is broken by the non-collinear coplanar order within a centrosymmetric crystal structure and TRS in momentum space is preserved even in the presence of SOC.”*

Below, I have enumerated several questions and comments that I would like the authors to address before I make a final decision on this manuscript.

1. In the second paragraph, the authors try to compare their NREE in p-wave magnets with NREE in other systems and traditional relativistic EE. The key point they keep emphasizing is the time-reversal symmetry (TRS). So here are several questions:

(1) To me, TRS is not directly related to the Edelstein effect, which only requires the broken of inversion symmetry. Then why do we need to care about the broken or preservation of TRS in different systems?

Reply: We thank the reviewer for raising this insightful question. The conventional EE was introduced for Rashba systems in non-centrosymmetric crystals, where spin-orbit coupling (SOC) induces an anti-symmetric spin texture in momentum space. More recently, it has been proposed that the non-collinear magnets can also enable charge-to-spin conversion without relying on SOC though these systems deviate from the time-reversal preserving nature observed in Rashba systems. The observation of spin-charge conversion of the Edelstein type has only been achieved directly in non-magnetic systems because typically, in a system whose band structure breaks time reversal symmetry, one often has signals that would inevitably mask this effect - e.g. anomalous Hall effect transport effects, spin-injection accumulation, etc.

In our work, we demonstrate that coplanar p-wave magnets exhibit a complete exchange-driven analogue of SOC within a non-relativistic regime. Remarkably, these TRS preserving systems generate a unique, uni-directional out-of-plane spin response a phenomenon that, to our knowledge, has not been explored before and that can serve as an important measurement of p-wave magnets themselves.

In response to point (1), we emphasize that, along with broken inversion, the modulation of spin-polarization is also a crucial requirement to get finite EE (section 2 of SI). In Rashba systems, this modulation appears as a directional change in the spin-texture in momentum space, whereas in p-wave systems, it manifests as a modulation in the magnitude of momentum space spin polarization a distinction illustrated in Fig.1 of the main manuscript.

We also note (also in connection to other comments below), that the fact that p-wave magnets have time reversal as a point group symmetry operation, also means that even in the presence of spin-orbit coupling, the band structure remains time reversal symmetric.

Action taken: To address the importance of preservation of TRS in charge-to-spin conversion we have included the following statements in the Introduction (same paragraph as above, re-written here for completeness): *“The EE is, in principle, allowed if the system breaks inversion symmetry irrespective of the origin of the response, be it SOC or non-collinear order. The conventional relativistic Rashba-Edelstein effect in TRS systems originates from SOC induced anti-symmetric spin textures in momentum space. The*

recently predicted *p*-wave magnets (Fig. 1c) have TRS as a point group operation, thus exhibiting TRS in momentum space akin to Rashba systems. However, unlike Rashba spin splitting, which, is isotropic, our *p*-wave spin splitting is strongly anisotropic and momentum dependent [35]. This can give rise to out-of-plane non-equilibrium spin accumulation, a feature not permitted in the usual relativistic Rashba-Edelstein effect. This *p*-wave magnet NREE contrasts with recent reports [33, 34] that obtain a NREE in non-collinear magnetic order in non-centrosymmetric crystals and which do not have TRS as a point group symmetry. In *p*-wave magnets, the inversion symmetry is broken by the non-collinear coplanar order within a centrosymmetric crystal structure and TRS in momentum space is preserved even in the presence of SOC. ”

(2) What is explicitly the TRS and TRS breaking in momentum space?

Reply: TRS in momentum space for a magnetically ordered system (which breaks TRS in direct space) requires \mathcal{T} to be a point group symmetry of the ordered ground state. This $\mathcal{T}\vec{t}$ symmetry in a collinear system also leads to spin degeneracy in the non-relativistic band structure. In a non-collinear system spin degeneracy is not protected and can lead to a *p*-wave magnet with antisymmetric spin-splitting.

A *p*-wave magnet (*p*-wave symmetry spin-splitting in momentum space) is defined by having $\mathcal{T}\vec{t}$, broken parity, and a spin symmetry group enforcing a single nodal surface. We further consider candidates with co-planar non-collinear order in the direct crystal space, since they have the unique property of the collinear out-of-plane spin order in the momentum space in the non-relativistic limit.

The point group TRS element has the consequence that the spin-splitting is antisymmetric in momentum space:

$$E_{\mathbf{k}}(\boldsymbol{\sigma}, \mathbf{k}) = E_{\mathbf{k}}(-\boldsymbol{\sigma}, -\mathbf{k}) . \quad (3)$$

It is important to note that this also implies that even in the presence of SOC, the system will maintain this property. This is of course the case since the SOC Hamiltonian does not break this point group symmetry and therefore the spin-splitting arising from it will remain antisymmetric.

If the system does not have TRS as a point group symmetry element, this relation (Eq. (3)) is no longer generally valid, even though in the non-relativistic limit the relation may hold. Specifically, a general feature of co-planar magnetic order in the non-relativistic limit is that the out-of-plane spin component is antisymmetric and the in-plane components are symmetric (see also the next response below). In this non-relativistic limit there can be cases where additional spin-symmetries make the in-plane spin component in momentum space zero for the co-planar magnetic ordered systems. This then leads to a pure out-of-plane antisymmetric spin-splitting. As result, the above Eq. (3) is preserved by these symmetries in this limit. However, in the presence of SOC this is no longer valid, i.e., $E_{\mathbf{k}}(\boldsymbol{\sigma}, \mathbf{k}) \neq E_{\mathbf{k}}(-\boldsymbol{\sigma}, -\mathbf{k})$ in general. This is for example the case for systems already reported in the literature, for example, in ref. 31 [Phys. Rev.

B 101, 220403 (2020)] of our manuscript. Here the systems will not have generally anti-symmetric TRS spin-splitting in the presence of SOC and, in fact, the system will generally allow for TRS odd responses, which would be forbidden by symmetry in the p-wave magnets.

Action taken: To define the action of time-reversal symmetry in momentum space, we have included the statement *“By virtue of \mathcal{T} being an individual symmetry operation of the point group both in the presence and the absence of SOC, the momentum space energy dispersion satisfies the criteria: $E(\mathbf{k}, \boldsymbol{\sigma}) = E(-\mathbf{k}, -\boldsymbol{\sigma})$, where $\boldsymbol{\sigma}$ is the spin polarization and \mathbf{k} is the momentum.”* in the ‘Introduction’ (page 3).

(3) The TRS they are talking about in p-wave magnets in momentum space is actually not the real TRS, but an alternative one from Tt (t is transition) in the real space. More than just Tt, a system can break TRS but preserve TU, TUt (U is the proper spin rotation) in the real space but still have alternative TRS in the momentum space. So from this point of view, the papers they cite to claim that other NREE systems have no TRS are actually do have alternative TRS in the momentum space, which is just the same as their work here.

Reply: We thank the reviewer for raising this point, which gives us the opportunity to explain in more detail the presence of TRS in the p-wave systems from symmetry analysis and make a comparison with other systems.

To explain, we chose the material candidate CeNiAsO which is also considered in the manuscript to calculate NREE. The system in its non-collinear spin-ordered state obeys 2.1’ point group containing symmetry operations: $\{E, C_{2y}, \mathcal{T}, C_{2y}\mathcal{T}\}$. The magnetic point group explicitly contain the time reversal (\mathcal{T}) as an independent symmetry operator. The articles we cited reporting NREE previously do not contain the time reversal as a symmetry operation. For example, we present here the symmetry table for non-collinear magnetic orders of LuFeO₃ (ref. 33) and Mn₃IrSi (ref. 34). The non-collinear co-planar spin configuration of LuFeO₃ obeys 6m’ magnetic point group containing symmetry operations: $\{E, C_{6z}, C_{3z}, C_{2z}, -C_{3z}, -C_{6z}, M_x\mathcal{T}, M_y\mathcal{T}, M_1\mathcal{T}, M_{xy}\mathcal{T}, M_2\mathcal{T}, M_3\mathcal{T}\}$, where $M_{1,2,3}$ are three diagonal planes of hexagon. This absence of TRS is immediately visible from the anisotropic non-antisymmetric spin splitting of LuFeO₃ as presented in the colormap of S_z in Fig. 3c of ref. 33 *Nat. Commun.* **15**, 7663 (2024). Similarly, we present the symmetry operation of the non-collinear, non-coplanar magnetic order of Mn₃IrSi as studied in ref. 34. The relevant magnetic point group of Mn₃IrSi is 23.1 containing symmetry operations $\{E, C_{2x}, C_{2y}, C_{2z}, \pm C_{3xyz}, \pm C_{3xy-z}, \pm C_{3-xyz}, \pm C_{3x-yz}\}$.

In the non-relativistic limit, where spin symmetries apply (spin symmetries have generally distinct operations in spin and momentum space), the $\mathcal{T}\vec{t}$ has also important consequences. In a coplanar non-collinear system within the non-relativistic systems the spin symmetry $[C_{2\perp}\mathcal{T}||\mathcal{T}]$ (where \perp is the perpendicular axis

to the plane of the coplanar spins) enforces that in momentum space the out-of-plane-component of the spin is antisymmetric and the in-plane component of the spin is symmetric. That is, in the non-relativistic limit for a coplanar non-collinear order $S_{\perp}(\boldsymbol{\sigma}, \mathbf{k}) = -S_{\perp}(-\boldsymbol{\sigma}, -\mathbf{k})$ and $S_{\parallel}(\boldsymbol{\sigma}, \mathbf{k}) = S_{\parallel}(-\boldsymbol{\sigma}, -\mathbf{k})$. In the p-wave magnet candidate CeNiAsO with $\mathcal{T}t$, this implies that $[C_{2\perp}||\vec{t}]$ is a symmetry, making the $S_{\parallel} = 0$ and therefore enforcing the antisymmetric splitting.

In the non-relativistic limit, as the referee also points out, other spin symmetries (referred as $\mathcal{T}U$ by the referee) can, in principle, make the in-plane spin component in momentum space vanish. This fact is already established, for example, in ref. 31 [*Phys. Rev. B* **101**, 220403 (2020)], where the three criteria were proposed to get anti-symmetric spin splitting are (1) a triangular unit with an AFM structure, (2) inversion symmetry breaking, and (3) active magnetic toroidal multipoles (imaginary hopping) in the model Hamiltonian. We have also explored this in our follow-up preprint Nayra A. Alvarez Pari. et. al [arXiv:2412.10984](https://arxiv.org/abs/2412.10984) (2025). However, when SOC is taken into account (in particular with systems such as the Eu compound with strong SOC), these systems no longer preserve this anti-symmetric spin-splitting, i.e. the band structure breaks TRS, whereas the p-wave magnets will retain the TRS with or without SOC. Specifically, in the coplanar helical phase of EuIn₂As₂, the $[C_{\pm 3z}||E]$ symmetry along with coplanar symmetry $[C_{2\perp}\mathcal{T}||\mathcal{T}]$ gives rise to non-relativistic anti-symmetric spin splitting over momentum space. However, in the presence of SOC this is no longer the case, since the SOC single particle Hamiltonian does not possess these extra spin symmetries.

We also note that in these systems that break the point group TRS, according to the Neumann principle, TRS broken responses will be generally allowed. This will not be the case for the p-wave magnets.

Action taken: To discuss the context of anti-symmetric spin polarization for TRS broken systems we have included the following sentences on page 9 of the revised manuscript: *“We plot the band dispersion of CeNiAsO in Fig.4e. The bands have opposite out-of-plane spin polarization S_z for opposite momentum with a linear crossing at the Γ point. The anti-symmetric S_z projection is a direct consequence of \mathcal{T} being a point group symmetry operation for CeNiAsO, which also restricts the responses to be time-reversal-even even in the presence of SOC. We note that there can be some co-planar magnetic ordered systems that do not have TRS as a point group symmetry element but have extra spin-symmetries that lead to the in-plane spin-component in the non-relativistic band structure to become zero, as in e.g. ref.[31]. This leads to an antisymmetric out-of-plane spin splitting in the non-relativistic band structure. However, in the presence of SOC, these systems that lack TRS as point group symmetry no longer have an antisymmetric spin-splitting band structure and, in fact, allow for time-reversal-odd responses as well.”*

2. The authors claim that the response, or more specifically, susceptibility, of their NREE in p-wave magnet is several times larger than in other systems, which is the main advantage of their NREE over other systems.

RFig. 3. The non-relativistic Edelstein response (χ_S) for helical phase of EuIn_2As_2 (red) and p -wave CeNiAsO (blue) respectively. For CeNiAsO the NREE response tensor is finite at Fermi energy (black dashed vertical line) with value of $\chi_S = -0.05\hbar/V\text{\AA}^2$.

However, In the work published by Nayra A. Álvarez Pari et al. arXiv:2412.10984, they calculated a one times larger susceptibility in the non-collinear EuIn_2As_2 system than in CeNiAsO . What is the reason for that?

Reply: We thank the reviewer for noticing and acknowledging our follow-up work arXiv:2412.10984 on EuIn_2As_2 . The motivation for studying EuIn_2As_2 stems from the ongoing debate on the nature of the ground state of this fascinating material. Although these systems have a strong SOC component, we show in this work that the characteristics of the Edelstein effect can be exploited to distinguish the phases. Here, similarly to our case, the magnetic order breaks the parity allowing for the effect, but unlike in the p -wave case, the time-reversal symmetry is not a point group symmetry of the system.

In this context, we identified the possible helical phase of EuIn_2As_2 as a promising platform for anisotropic NREE, in contrast to previously reported non-collinear systems featuring both in-plane and out-of-plane NREE. Although the maximum amplitude of NREE in EuIn_2As_2 is of the order of p -wave candidate CeNiAsO , EuIn_2As_2 is semiconducting. The need for strong doping of the EuIn_2As_2 to experimentally obtain the charge to spin accumulation will alter the band structure and it is unlikely to reach high metallicity, reducing its efficiency as compared to the metallic p -wave candidate CeNiAsO . Thus, the p -wave CeNiAsO remains a more promising candidate for achieving the giant anisotropic NREE, potentially surpassing the values reported so far by a significant margin.

In conclusion, while the concept of non-relativistic Edelstein effect is not entirely new, the p -wave magnet has not yet been used as a research system for NREE. But apart from that, this work seems to lack enough novelty to be published in *Nat. Commun.* Moreover, the advantage they emphasize of the large susceptibility

response in CeNiAsO might be larger than normal EE, but is indeed not that surprisingly large in the NREE region.

Reply: We sincerely appreciate the reviewers insight in recognizing that our work is the first to report this giant anisotropic NREE response of p -wave magnets. This phenomenon arises from the unique odd-parity and inherent $\mathcal{T}t$ -symmetric spin polarization. Unlike recent studies on NREE in non-collinear magnets, where the point group symmetry elements \mathcal{T} and P are absent, our work presents a fundamentally different perspective, highlighting the potential for spintronic applications of anisotropic out-of-plane responses that are direct analogs to SOC but arising from exchange interaction and having a much higher efficiency of charge-to-spin conversion.

Action taken: We have included the following sentences in the ‘Discussion and Outlook’ section to emphasize the importance and uniqueness of our findings with respect to the already reported literature: *“This type of coplanar noncollinear magnetic order preserves TRS in momentum space (through the $\mathcal{T}\vec{t}$ symmetry) while exhibiting odd-parity polarized band structure. Our p -wave NREE inherits the distinctive features of the unconventional p -wave magnets and is thus distinct from the conventional relativistic Rashba-Edelstein effect in three aspects: First, the spin symmetry $[C_{2\perp}||\vec{t}]$ of p -wave magnets forces the polarization direction in the band structure to be perpendicular to the plane of spin coplanarity. This can give rise to out-of-plane nonequilibrium spin accumulation, a feature not permitted in the usual relativistic Rashba-Edelstein effect and in other NREEs in noncoplanar magnets. Second, the unique anisotropic momentum-dependent splitting in p -wave magnets gives them an anisotropic nodal EE signature which is absent in the relativistic but also other magnetic system counterparts. Third, the non-equilibrium spin accumulation within p -wave magnets does not originate from the SOC effect but rather from the exchange interaction of the non-collinear magnetic order. In turn, NREE in p -wave magnets can lead to unprecedented high efficiency in spin-to-charge conversion.”*

We hope the reviewer finds the revised manuscript clearer and shares our excitement about this work.

We thank the reviewers for carefully reviewing our manuscript and for supporting our work. Their feedback has helped us to improve the readability of our manuscript and to make it clearer. Below, we address the reviewers comments. The reviewers comments/questions are in blue colored font, and our responses are in black colored font. All the changes in this revised manuscript are marked in red colored font.

Reviewer #2:

The authors answered my questions properly. I suggest the publication of this paper in Nature Comm.

Reply: We sincerely thank the reviewer for the kind recommendation for the publication of our article.

Reviewer #3:

The authors have properly addressed most of my questions and modified the manuscript accordingly. However, there is still several points that need to be addressed before I can recommend this work for publication:

Reply: We thank the referee for the thoughtful evaluation and for appreciating our response. Below, we address all the further questions raised by the referee.

1. While it is acknowledged that Tt (time-reversal T combined with a translation t operation) possesses certain advantages, this does not imply that TU (U is a proper spin rotation) is inherently deficient. Consider a scenario where the effective time reversal operation of the system is TU rather than Tt . When the introduced spin-orbit coupling (SOC) is sufficiently small, the presence of spin splitting can still be sustained, thereby permitting the occurrence of NREE. Conversely, when the SOC becomes sufficiently large, the existence of in-plane spin components even in the case of a p-wave magnet (pM) results in a reduction of spin accumulation, that is, the response associated with NREE, due to these in-plane components. It seems to me that there is no remarkable difference between TU (also TUt) and Tt if SOC is included. Is it true? Moreover, if the analysis is confined to spin accumulation along the z-direction, MzT can produce an effect equivalent to that of Tt , correct?

Reply: We thank the reviewer for recognizing and appreciating the advantages of nodal p -wave magnets with Tt symmetry over time reversal symmetry (TRS) broken magnets with a \mathcal{TU} spin symmetry (here a \mathcal{TU} system is a coplanar compensated magnet with a \mathcal{TU} spin symmetry in the non-relativistic limit which enforces the in-plane component to vanish but has no $\mathcal{T}t$ symmetry), as well as other classes of non-collinear magnets. As we stated in our previous reply, the presence of spin-orbit coupling (SOC) generates nonzero

in-plane components of spin expectation values for both $\mathcal{T}t$ and $\mathcal{T}U$, but with generally different symmetries. While SOC generally reduces the efficiency of the NREE in both cases, there is a key fundamental remaining remarkable difference between the Tt system and the TU system in the presence of SOC. This can be seen in both the ground state spin texture characteristics and the responses to external fields:

- **Ground state characteristics:** In the presence of SOC, the TRS-broken $\mathcal{T}U$ systems will generally have different magnitudes of the spin components for opposite momenta states, i.e. $|S_i(\mathbf{k})| \neq |S_i(-\mathbf{k})|$ for $i = x, y, z$, since the system breaks \mathcal{T} in the point group and the $\mathcal{T}U$ spin symmetry is no longer present. This is markedly different from the nodal p -wave magnets that we consider, where the spin texture is purely antisymmetric with opposite momenta, i.e. $S_i(\mathbf{k}) = -S_i(-\mathbf{k})$ for $i = x, y, z$, due to the presence of \mathcal{T} symmetry in the point group, irrespective of the strength of the SOC.
- **Responses:** In the presence of strong SOC, for the TRS-broken $\mathcal{T}U$ systems, along with the generation of an asymmetric in-plane relativistic spin accumulation, there will be generally allowed TRS-broken responses, such as, e.g., anomalous Hall effect (if the magnetic point group is a ferromagnetic point group). This allowed TRS-broken symmetry responses in the $\mathcal{T}U$ systems will increase with the strength of the SOC and can suppress or, in principle, can surpass the time reversal symmetric out-of-plane Edelstein response. In contrast, any TRS-odd responses will be identically zero for the nodal p -wave magnets that we consider due to the explicit presence of \mathcal{T} in the point group, even in the presence of a strong SOC.

We consider these to be very distinct and experimentally verifiable remarkable differences between $\mathcal{T}U$ (also $\mathcal{T}Ut$) and $\mathcal{T}t$ systems when SOC is included.

The presence of any $M_i\mathcal{T}$ (where $i \in x, y, z$) symmetry enforces the spin component perpendicular to the mirror plane to be anti-symmetric, and those parallel to the mirror plane to be symmetric, under application of anti-unitary mirror. In coplanar nodal p -wave magnets, the emergence of only out-of-plane spin-polarization is constrained by $[C_{2z}||\vec{t}]$ or, equivalently, $\mathcal{T}t$ symmetry, as mentioned in our manuscript. In contrast, the presence of e.g. $M_z\mathcal{T}$ alone permits symmetric S_x and S_y components for $(k_x, k_y, k_z) \rightarrow (-k_x, -k_y, k_z)$. Additionally, the $M_i\mathcal{T}$ does not connect the opposite momentum pairs i.e. \vec{k} and $-\vec{k}$. Thus, $M_z\mathcal{T}$ does not serve the same role as $\mathcal{T}t$, which – as emphasized above – enforces the in-plane spin component in the band structure to be zero.

2. I also want to address an issue concerning p -wave magnets. The author asserted that a pM must possess a nodal plane; however, there should exist nodeless pMs , corresponding to the chiral p -wave superconductor in superconductivity. This is exemplified by the beta phase pM , as noted in arXiv:2504.14577v1.

Reply: We thank the reviewer for noting this point. We want to emphasize that the focus of the present manuscript is restricted to nodal p -wave magnets (α -type), whose symmetry enforces an out of plane collinear spin polarization of the electronic band structure, which yields the unique anisotropic out-of-plane nonrelativistic Edelstein response (NREE). Indeed, there exist nodeless p -wave magnets (β -phase) as discussed in our preprint Hellenes et. al. arXiv:2309.01607v1 and also in the preprint arXiv:2504.14577v1 mentioned by the reviewer. However, for the nodeless non-coplanar p -wave magnets, such as is the case in Ce_3InN (considered in arXiv:2309.01607v1), the non-relativistic spin-polarization of the electronic band structure as well as nonequilibrium spin-accumulation is coplanar in contrast to the unique anisotropic collinear out-of-plane NREE response observed in nodal p -wave magnets that we consider. Also, the non-collinear co-planar spin texture in these β -phase nodeless magnets will make the efficiency lower and difficult to distinguish by symmetry from SOC induced contributions.

Action taken: Following the reviewers comment, to make clear the focus on nodal p -wave magnets, we have now changed the title from “*Highly Efficient Non-relativistic Edelstein effect in p -wave magnets*” to “*Highly Efficient Non-relativistic Edelstein effect in nodal p -wave magnets*”. To emphasize the point that our focus of this study is nodal p -wave magnets, we have included the following sentences in the introduction of the revised manuscript: “*There exist examples of non-coplanar p -wave magnetic systems which can feature nodeless co-planar anti-symmetric spin polarization. However, for the present study, we are interested in nodal p -wave magnets candidates with coplanar spin order in position space showing this unique collinear out-of-plane non-relativistic spin polarization in momentum space.*”

3. I am also curious about the rigorous definition of P -wave magnet. Does it need to forbid all the even-order polynomials of the $k \cdot p$ Hamiltonian? For example if the high-symmetry point has D_2 magnetic group symmetry, then even-order polynomials are also forbidden. Then will it be a p -wave magnets even considering SOC? I would highly recommend that the authors provide a thorough definition and symmetry requirement of p -wave magnets.

Reply: We define the nodal p -wave magnets as a coplanar compensated spin-ordered magnet with $\mathcal{T}t$ symmetry and with a single symmetry-enforced nodal surface, as discussed in arXiv:2309.01607. This rigorous definition imposes all the characteristics of the nodal p -wave magnet that we have already discussed, such as, e.g., antisymmetric spin polarization with momentum reversal.

The symmetry requirements for odd parity magnets (nodal or nodeless) in general (as also mentioned in the main manuscript) are: 1. It must be a non-collinear magnet, 2. The real space inversion symmetry must be broken, 3. PT symmetry must be broken, and 4. the magnet possesses $\mathcal{T}t$ symmetry.

Due to the presence of explicit TRS in the point group, even in the presence of SOC, all the even-order

polynomials of the $k \cdot p$ model Hamiltonian will vanish for a pure p -wave magnet.

The mentioned D_2 magnetic point group, namely the 222.1, includes $\{E, C_{2x}, C_{2y}, C_{2z}\}$ and does not contain \mathcal{T} explicitly as a point group symmetry operation, which is the key criterion for p -wave magnets. The rotational symmetry operations imply that if two momentum points are related by, say, C_{2x} rotation, then the spin components perpendicular to the rotation axis, namely S_y and S_z , will exhibit opposite signs at those momentum points. To verify this behavior, we performed calculations on a candidate material, FePO_4 , which hosts a non-collinear spin configuration, as shown in RFig1-a, consistent with the D_2 magnetic point group. Due to the rotational symmetry criteria mentioned above, we see the x -component of the spin projection along the $\bar{X} (-0.5, 0, 0) - \Gamma - X (0.5, 0, 0)$ path is antisymmetric (see RFig1-b). However, these rotational symmetries do not ensure universally antisymmetric spin polarization across the Brillouin zone. For instance, at the point $P : (0.25, 0.25, 0.25)$, the corresponding opposite momentum pairs are not connected by any of the D_2 rotation symmetries. Along the $\bar{P} - \Gamma - P$ path, we observe that the S_z component is anti-symmetric (see RFig1-d), but the S_y component is symmetric, as marked by the oval in RFig1-c.

RFig. 1. (a) Non-collinear magnetic order of FePO_4 obeying 222.1 magnetic point group. The x - (b), y - (c) and z - (d) components of spin are plotted on top of the total band structure along high-symmetry paths.

In contrast, for all odd-parity wave magnets where the magnetic point group includes explicitly TRS, \mathcal{T} , states with opposite momenta have opposite spins. This distinct behavior highlights the fundamental role of $\mathcal{T}t$ symmetry in shaping antisymmetric spin textures that we consider in the nodal p -wave magnets.

We hope that the reviewer will now be satisfied with our thorough and detailed response, together with the improved version of our manuscript.